# On the nature of Co$_n^{±/0}$ clusters reacting with water and oxygen
Lijun Geng[1,5], Pengju Wang[2,5], Shiquan Lin[1,3], Ruili Shi[2], Jijun Zhao[2,4] ✉ & Zhixun Luo [1,3] ✉

Bulk cobalt does not react with water at room temperature, but cobalt nanometals could yield corrosion at ambient conditions. Insights into the cobalt cluster reactions with water and oxygen enable us to better understand the interface reactivity of such nanometals. Here we report a comprehensive study on the gas-phase reactions of Co$_n^{±/0}$ clusters with water and oxygen. All these Co$_n^{±/0}$ clusters were found to react with oxygen, but only anionic cobalt clusters give rise to water dissociation whereas the cationic and neutral ones are limited to water adsorption. We elucidate the influences of charge states, bonding modes and dehydrogenation mechanism of water on typical cobalt clusters. It is unveiled that the additional electron of anionic Co$_n^-$ clusters is not beneficial to H$_2$O adsorption, but allows for thermodynamics- and kinetics-favourable H atom transfer and dehydrogenation reactions. Apart from the charge effect, size effect and spin effect play a subtle role in the reaction process. The synergy of multiple metal sites in Co$_n^-$ clusters reduces the energy barrier of the rate-limiting step enabling hydrogen release. This finding of water dissociation on cobalt clusters put forward new connotations on the activity series of metals, providing new insights into the corrosion mechanism of cobalt nanometals.

As one of the three ferromagnetic metals in the periodic table of elements, cobalt is widely used in magnetic alloys with the advantage of heat resistance. Cobalt-based materials manifest a wide range of applications including permanent magnets[1], information storage[2] and aerospace manufacturing[3]. Since air and water affect the lifetime of these materials and the retention of their properties[4,5], corrosion is an ever-present concern in the world of metals[6–8]. It is important to fully understand the interface interactions and reaction mechanism, which can guide the rational design of anticorrosion strategy for practical applications. The related metal–water interactions are also an important theme of research in chemistry and biology as well as energy source and environment[9,10].

On the other hand, the low-cost and high-efficiency hydrogen production by water dissociation via electrolysis and photocatalysis is a long-term research topic[11–14], for which cobalt nanocatalysts have attracted extensive interest, although cobalt usually does not react with water at ambient conditions[15–21]. Catalytic O–H dissociation and HAT is vital to water dehydrogenation and O-O bond formation[22–24]; however,

catalytic oxidative dehydrogenation often exhibits limited activity and poor selectivity, despite decades of research efforts in this field. Small metal clusters possess distinct catalysis in contrast to their bulk analogues due to the quantum size effect and unique electronic structures[25]. For instance, dehydrogenation of water on some Al$_n^-$ clusters was observed at room temperature[26–28], leading to the establishment of a complementary active site (CAS) mechanism[27]. Dehydrogenation of H$_2$O molecules by reacting with gas-phase vanadium clusters was also noted[29], showing diverse V$_n$O$^+$, V$_n$O$_2^+$ and V$_n$O$_3^+$ products by rapid reactions of V$_{n\geq3}^+$ with water in a fishing mode[30]. In contrast to aluminium and vanadium, however, cobalt does not support water dehydrogenation according to the activity series of metals[31]. There comes a pending question if sub-nanometer cobalt clusters can support spontaneous water dehydrogenation and oxidation, which leads to a better understanding of the corrosion of cobalt nanosurfaces.

Based on this motivation, herein we report a comprehensive study of the gas-phase reactions of Co$_n^{±/0}$ clusters with water and oxygen.

[1]Beijing National Laboratory for Molecular Sciences (BNLMS), State Key Laboratory for Structural Chemistry of Unstable and Stable Species, Institute of Chemistry, Chinese Academy of Sciences, Beijing, P. R. China. [2]Key Laboratory of Materials Modification by Laser, Ion and Electron Beams, Ministry of Education, Dalian University of Technology, Dalian, P. R. China. [3]University of Chinese Academy of Sciences, Beijing, P. R. China. [4]Guangdong Basic Research Centre of Excellence for Structure and Fundamental Interactions of Matter, Guangdong Provincial Key Laboratory of Quantum Engineering and Quantum Materials, School of Physics, South China Normal University, Guangzhou, P. R. China. [5]These authors contributed equally: Lijun Geng, Pengju Wang. ✉e-mail: zhaojj@scnu.edu.cn; zxluo@iccas.ac.cn

Well-resolved $Co_n^{\pm/0}$ (c.a., $n = 1–30$) clusters are prepared, and their reactions with water are studied by using our self-developed ultrafast deep ultraviolet laser ionisation mass spectrometer (DUV-LIMS, Supplementary Fig. S1)[32,33]. As a result, we found all these $Co_n^{\pm/0}$ clusters react with oxygen to form diverse oxides. However, the $Co_n^+$ clusters readily react with water giving rise to diverse adsorption products, which contrasts with the anionic $Co_n^-$ clusters which allow for dehydrogenation in reacting with water. Combined with density functional theory (DFT) calculations, we illustrated the reaction dynamics and unveiled the altered binding mode of water on the small $Co_n^-$ cluster anions ($Co_n^-\cdots H–OH$) compared with their cationic and neutral analogues ($Co_n^{+/0}\cdot OH_2$). Apart from the charge effect, we also elucidated the spin effect and cooperative multi-site effect that promote water dissociation and dehydrogenation on the $Co_n^-$ clusters, showing enhanced activity of such cobalt clusters without being restricted by the principles of activity series of metals.

## Results and discussion

### Anionic $Co_n^-$ clusters reacting with water

The reactions of $Co_n^{\pm/0}$ clusters with oxygen have been addressed in our previous study[34], showing a tendency to form diverse oxides (Supplementary Fig. S2) but with a stable cluster $Co_{13}O_8$ showing up in the presence of sufficient oxygen reactant. Here we emphasize on the reactions with water. Figure 1A presents a typical mass spectrum of anionic $Co_n^-$ ($n = 5–59$) clusters in the absence and presence of water carried by bubbling of He buffer gas. The prepared $Co_n^-$ clusters display a regular Gaussian/Rayleigh distribution centred at $Co_{25}^-$. Isotope-labelled water, $H_2^{18}O$, was used to exclude the interference of trace amount of oxygen contamination (also, deuterium water $D_2O$ was also used to unambiguously identify the dehydrogenation products, Supplementary Fig. S3). Figure 1B displays an enlarged area to visualise the products of $Co_n^-$ ($n = 10–20$) clusters reaction with $H_2^{18}O$. The observation of a series of products $[Co_n^{18}O]^-$, $[Co_n^{18}O_2]^-$ and $[Co_n(^{18}OH)_2]^-$ suggests that the $Co_n^-$ clusters undergo dehydrogenation with one and two water molecules.

We monitored the reaction of $Co_n^-$ clusters with water at varying doses of the water controlled by the pulsed valve (Fig. 2 and Supplementary Fig. S4). An estimation of reaction rates is given in Supplementary Fig. S5. When the cobalt clusters reacted with small amounts of water, a series of $[Co_n^{18}O]^-$ products (accompanied by minor $Co_nO^-$ contamination) were observed in the mass spectra. As the dose of water was gradually increased, the $[Co_n(^{18}OH)_2]^-$ and $[Co_n^{18}O_2]^-$ products appeared in the mass spectra (and the nascent $Co_nO^-$ contamination peaks disappeared). When a large amount of water was involved in the reaction, a series of $[Co_n(^{18}OH)_2]^-$ products dominated the mass spectra. Interestingly, the adsorption products $[Co_nH_2^{18}O]^-$ and $[Co_n(H_2^{18}O)_2]^-$ were absent in the mass spectrometry observation, indicating that the HAT and dehydrogenation proceeded rapidly. This was also verified by the experiments based on deuterium water ($D_2O$, Supplementary Fig. S3). This experimental observation challenges the previously established principles that cobalt reacts with protonic acid (but not $H_2O$)[31] to form $H_2$. The dehydrogenation reactions of anionic cobalt clusters with water can be written as,

$$Co_n^- + H_2O \rightarrow Co_nO^- + H_2 \tag{1}$$

$$Co_n^- + 2H_2O \rightarrow Co_n(OH)_2^- + H_2 \tag{2}$$

$$Co_n^- + 2H_2O \rightarrow Co_nO_2^- + 2H_2 \tag{3}$$

### The reactions of neutral and cationic Co clusters

To compare the reaction behaviour of cobalt cluster anions, neutrals and cations, Fig. 3 displays the mass spectra of the cobalt cluster cations and neutrals before and after reacting with water, respectively. To distinguish the likely hydrogenation products, the isotope chemical $D_2O$ was used. As a

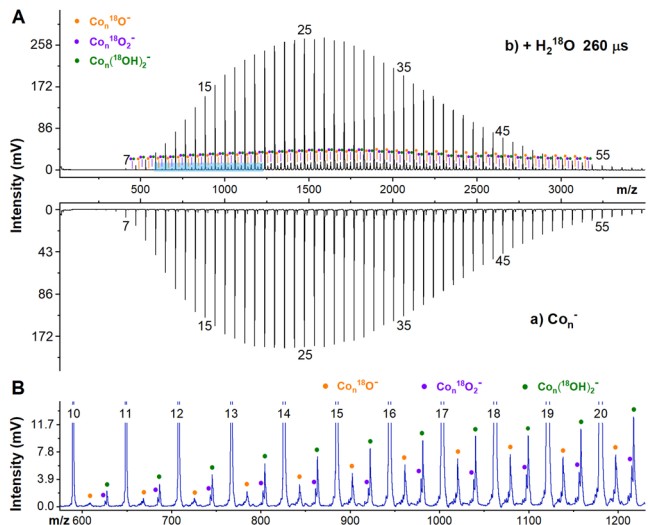

**Fig. 1 | Typical mass spectrum of $Co_n^-$ reacting with $H_2^{18}O$. A** Typical mass spectra of the $Co_n^-$ clusters produced by the homemade LaVa source, within a 35 mm nozzle and 10.0 atm He buffer gas, and the reaction products with $H_2^{18}O$ being introduced into the flow tube, controlled by a pulsed valve with a pulse width at 260 μs. **B** Enlarged area for the $Co_n^-$ ($n = 10–20$) clusters after the reaction with $H_2^{18}O$.

result, the cationic $Co_n^+$ clusters were found to adsorb multiple $D_2O$ molecules showing diverse $Co_n^+(D_2O)_m$ complexes; however, almost no dehydrogenation products were observed except for $Co_3^+$. The observation of strong water adsorption on $Co_n^+$ clusters is consistent with the previous studies of $Rh_n^+$ clusters reacting with water, as well as the $Co_n^+$ clusters reacting with $NH_3$ (ref. 35). Natural bond orbital (NBO) analysis shows maximal donor–acceptor orbital overlap interaction energy between the cationic clusters and $H_2O$ molecule (Supplementary Fig. S12), which is also in agreement with the results of charge decomposition analysis and potential scan for a $H_2O$ molecule in approaching a cobalt cluster (Supplementary Figs. S16 and S17). There is a similar case for the neutral cobalt clusters which also exhibit weak reactivity with water, with a few water-adsorption products being observed, such as $Co_{9-18}D_2O$ (Fig. 3B). The different reactions of $Co_n^{\pm/0}$ clusters with water embody the charge dependence of metal cluster reactivity, as revealed in the previous studies on the reactivities of Al, Nb and Rh clusters[36–39].

### Reaction dynamics and charge effect

We have conducted DFT calculations to elucidate the charge effect and $H_2$ release mechanism of cobalt clusters in reacting with water molecules. The structures of $Co_n^{\pm/0}$ and $[Co_nH_2O]^{\pm/0}$ ($n = 2–13$) clusters with different spin multiplicities are optimised at the PBE-D3/def2-TZVP level of theory (Supplementary Figs. S7–11 and Table S1). Interestingly, the lowest-energy structures of water adsorption on the cationic and neutral clusters prefer Co–O coordination (i.e., forming $[Co_nOH_2]^{+/0}$), with slight fluctuation of the bond lengths (Supplementary Fig. S14); however, water adsorption on the $Co_n^-$ ($n = 1, 2, 3, 5, 6$) clusters results in Co–H bonding ($Co_n^-\cdots H–OH$). This is consistent with the previous study of $M(H_2O)^-$ ($M = Cu, Ag, Au$)[40]. In addition, the binding energies ($E_{ad}$) of $H_2O$ onto the $Co_n^{\pm/0}$ clusters show significant charge dependence. The $E_{ad}$ values of the cations are larger than those of the neutral and anionic clusters (Supplementary Fig. S13), in line with the experimental observation that the cations can adsorb multiple water molecules while the reaction products of the anions are relatively small, although they support $H_2$ release.

We carried out DFT calculations on the thermodynamic energies and reaction kinetics typically for one and two $H_2O$ molecules to react with $Co_6^{\pm/0}$ which has an octahedral structure. As shown in Fig. 4, both $Co_6$ and $Co_6^+$ clusters suffer from unsurmountable energy barriers of the H-atom transfer (TS1_1 at 0.52 eV and 0.44 eV higher than the

**Fig. 2 | Mass spectrum of Co$_n^-$ reacting with H$_2^{18}$O of different doses. a** Mass spectra of the Co$_n^-$ clusters. **b–d** Mass spectra of Co$_n^-$ clusters after reactions with different amounts of H$_2^{18}$O, controlled by a pulsed valve with varying pulse widths at 180 μs, 240 μs, and 260 μs, respectively, corresponding to the original Supplementary Fig. S4. The peaks marked with stars (*) correspond to oxygen attachment due to the trace amount of contamination.

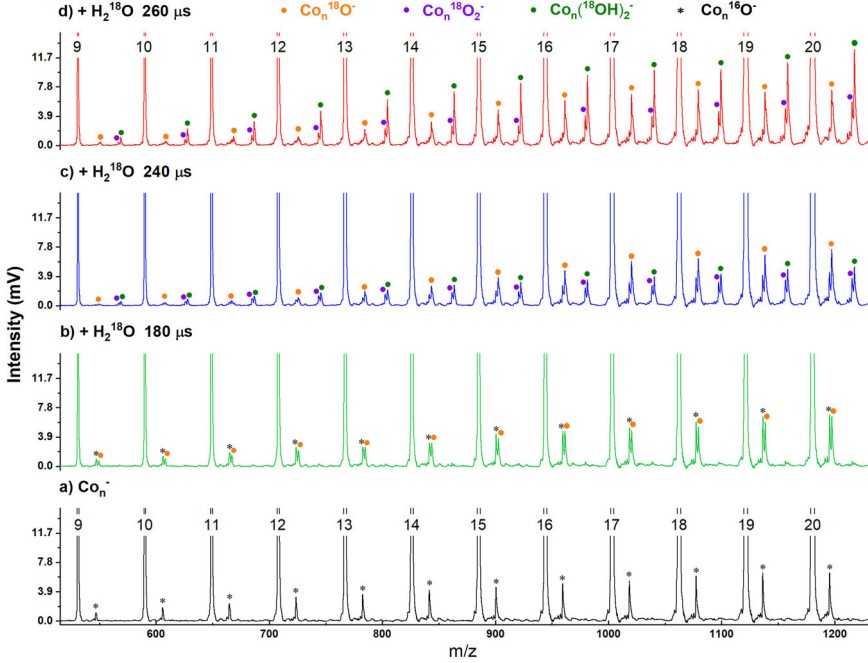

**Fig. 3 | Co$_n^{+,0}$ reacting with D$_2$O. A** Typical mass spectra of the cationic Co$_n^+$ (n = 2–15) clusters before and after reacting with D$_2$O in the flow tube. **B** Typical mass spectra of the neutral Co$_n$ (n = 2–30) clusters before and after reacting with D$_2$O.

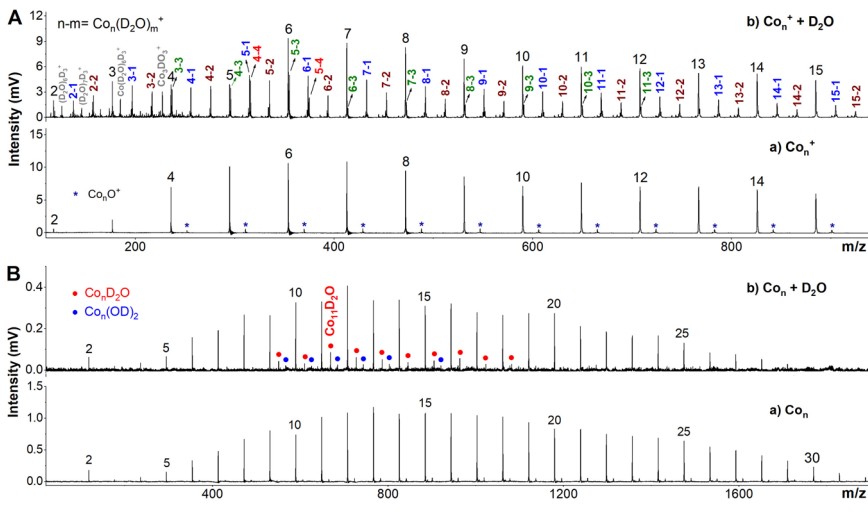

reactants); in contrast, HAT on the Co$_6^-$ cluster is thermodynamically favourable and kinetically favourable with a small energy barrier. While for the reaction of "Co$_6^{\pm/0}$ + 2 H$_2$O", the reaction coordinates indicate that three reactions are exothermic, but the energy for the H-atom transfer step (TS1_2) still differs from each other, and the neutral and cationic clusters take on larger single-step energy barriers. In addition, the reaction pathways for the cationic and anionic Co$_6^\pm$ clusters obey spin conservation, but the energy barrier of the rate-determining step for the Co$_6^-$ cluster (0.43 eV) is much lower than that of the Co$_6^+$ (1.88 eV). Also, for the reaction of neutral Co$_6$ with two H$_2$O molecules, the pathway of spin conservation (red-curve) suffers from a higher energy barrier of TS2_2 (1.52 eV); in comparison, the blue-curve pathway begins with a lower spin adsorption state ($^{13}$Co$_6$) and undergoes a relatively lower energy barrier (1.00 eV) of the rate-determining step. Although the spin crossing causes a reduced energy barrier, the dehydrogenation on neutral Co$_6$ is still not favourable compared with the anionic Co$_6^-$. It can be concluded that both spin states and charge effect play a dramatic role in the catalytic dehydrogenation on such nanometals[41]. Notably, the cool He

buffer gas could take away part of the energy during the reaction, rendering the cationic and neutral clusters not having enough energy for dehydrogenation (Supplementary Tables S2 and S3), especially for those having large energy barriers of the transition states.

## Size effect and multi-site cooperation

According to our DFT calculations, the reactions of Co$_3^-$ and Co$_{11}^-$ are also initiated by hydrogen-metal bonding adsorption (Fig. 5), similar to the aforementioned Co$_6^-$ in reacting with two H$_2$O molecules. Notably, the first H-atom transfer of a single H$_2$O on the Co$_3^-$ cluster is thermodynamically unfavourable. This is different from the previous finding of V$_{n\geq3}^-$ clusters in reacting with a single water molecule to release H$_2$, which is associated with the nature of the metal activity sequence. Nevertheless, Co$_3^-$ reacts with two H$_2$O molecules to release H$_2$, shedding light on the importance of synergetic active sites and multiple molecule cooperation.

In comparison, the reaction of Co$_{11}^-$ with a single H$_2$O finds a relatively larger energy gain of the adsorption and smaller energy barrier for the first H-atom transfer; nevertheless, the final transition state of

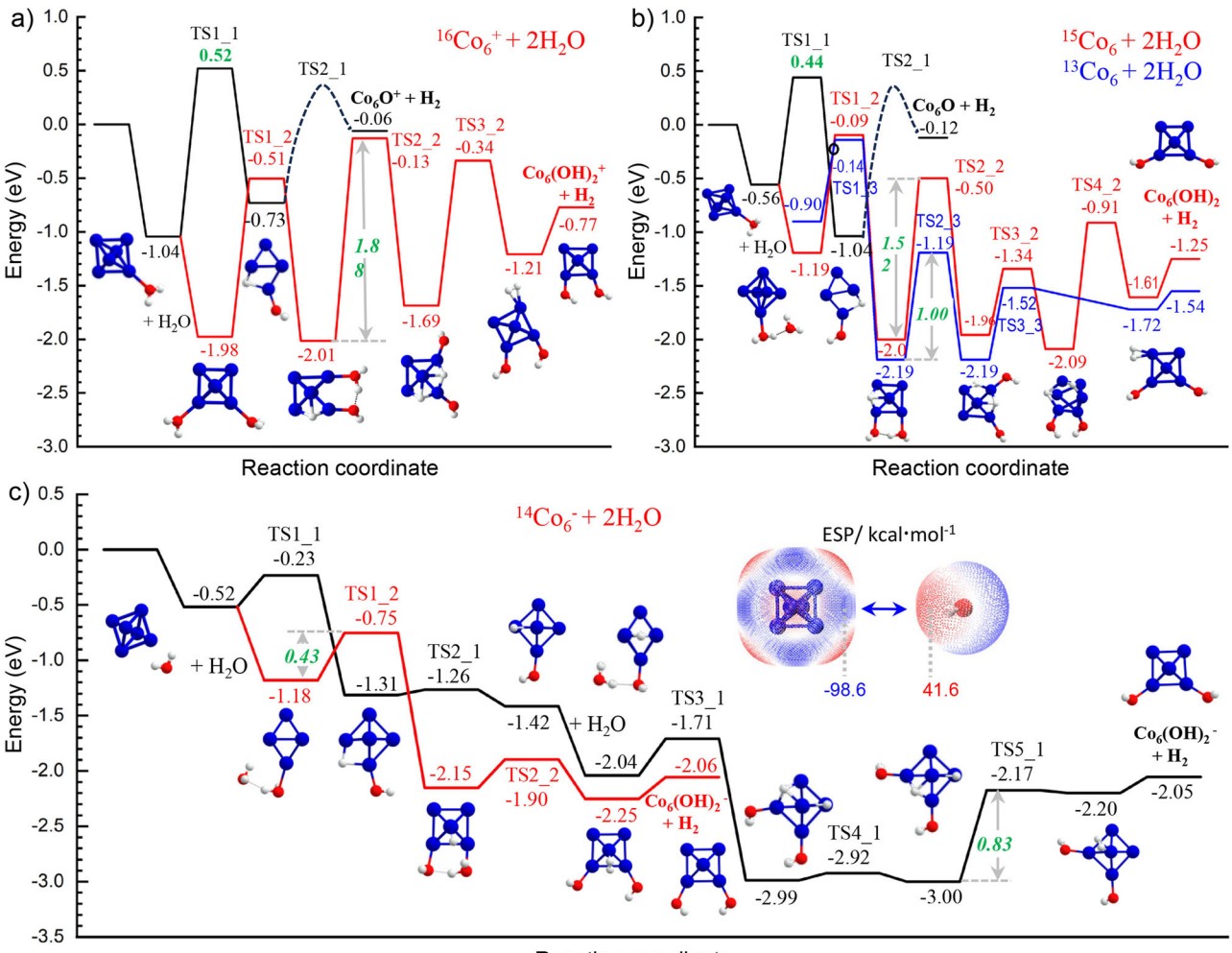

**Fig. 4 | Reaction coordinates of Co$_6$$^{\pm/0}$. a–c** The energy diagram for cationic $^{16}$Co$_6$$^+$, anionic $^{14}$Co$_6$$^-$, and neutral $^{15}$Co$_6$ clusters in reacting with one and two water molecules. Energies are given in eV. The inset shows electrostatic potentials (EPS, kcal mol$^{-1}$) for a H$_2$O molecule in approaching the Co$_6$$^-$ cluster with spontaneously regulated orientation.

H-H recombination for H$_2$ evolution displays comparable single-step energy barrier as the Co$_3$$^-$. Notably, the reaction of "Co$_{11}$$^-$ + 2 H$_2$O → Co$_{11}$(OH)$_2$$^-$ + H$_2$" shows a much smaller energy barrier (0.26 eV) for the H-atom transfer (TS1_2) compared with the rate-determining step for a complete dehydrogenation (1.34 eV for TS3_2, Supplementary Fig. S18). This coincides with the experimental observation of a larger mass abundance of Co$_n$(OH)$_2$$^-$ than Co$_n$O$_2$$^-$.

### The reactions of both water and oxygen
Considering that the corrosion of metals is essentially related to their chemical reaction with oxygen and water to form oxidation and dehydrogenation products, we further studied the reactions of the anionic Co$_n$$^-$ clusters by introducing both water and oxygen as reactants into the flow tube. The results are given in Fig. 6. It is seen that oxygen reacts with the Co$_n$$^-$ clusters to form Co$_n$O$_{2x}$$^-$ clusters without exception but with slightly lower reaction rates at Co$_5$$^-$ and Co$_6$$^-$, likely due to their structural stability[5]. Meanwhile, partial dehydrogenation products, including a series of [Co$_x$O$_y$·$^{18}$O]$^-$, [Co$_x$O$_y$·$^{18}$O$_2$]$^-$ and [Co$_x$O$_y$($^{18}$OH)$_2$]$^-$ were observed, indicating that oxygen undergoes competitive adsorption but does not hinder dehydrogenation. Nevertheless, from the diverse products of oxidation and dehydrogenation, it can be inferred that nanoscale cobalt suffers from inevitable corrosion, although this reactivity could not be so fast as iron. By referring to the corrosion equation of iron[42], the reactivity of cobalt clusters with water and oxygen could be summarised by an integrated reaction channel,

$$Co_n^- + xH_2O + yO_2 \rightarrow Co_nO_m(OH)_u(H_2O)_v^- + (x - v - u/2)H_2, x + 2y = m + u + v \tag{4}$$

We would like to put forward more discussion on the corrosion mechanism of nano-cobalt. Under ambient moist and oxygen-rich atmosphere, most metals suffer from spontaneous oxidation and likely thermodynamical dehydrogenation with few exceptions (gold and platinum)[43]. Some metals such as aluminium, chromium, magnesium and nickel can be well protected by a layer of impenetrable oxide coatings that prevents further destruction of the surface. So does the stainless steel which usually involves chromium and nickel to attain dense protection, thus avoiding unwanted corrosion. Whether or not a tight protective film, it is vital to avoid the formation of hydrated metal oxide and prevent the metal rusts from continually flaking off, without an exposure of fresh metal surfaces to oxygen and water. According to our results, it could be helpful to avoid negative charge accumulation; in other words, it would be important to check surface static charge regularly and keep neutral surfaces. In addition, a previous study found that Co$_{13}$O$_8$ is a highly stable cluster oxide[34], which could also breed a strategy of tight protective film like Al$_2$O$_3$.

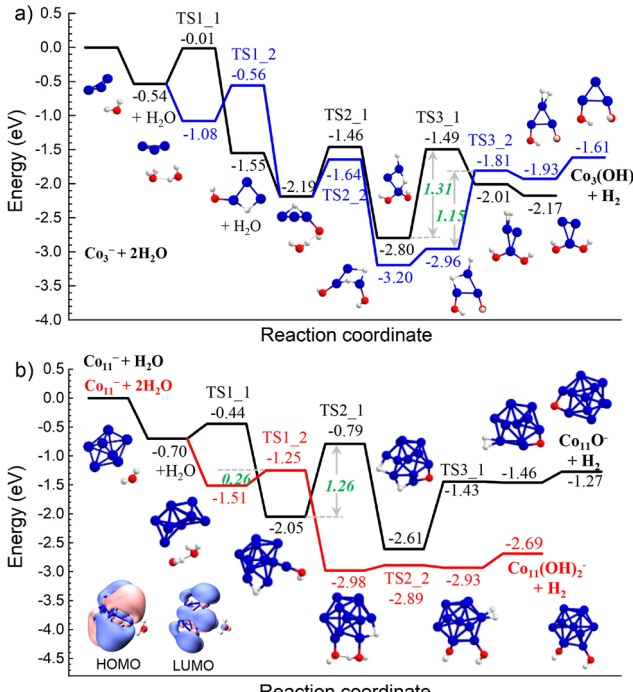

**Fig. 5 | A comparison of Co₃⁻ with Co₁₁⁻.** **a** Reaction energy diagram of "Co₃⁻ + 2 H₂O → Co₃(OH)₂⁻ + H₂". **b** Reaction energy diagram of "Co₁₁⁻ + H₂O → Co₁₁O⁻ + H₂" and "Co₁₁⁻ + 2 H₂O → Co₁₁(OH)₂⁻ + H₂". Energies are given in eV. The insets show the corresponding structures. The inset on the left bottom shows the HOMO and LUMO patterns of [Co₁₁H₂O]⁻.

## Conclusions

In summary, we report a joint experimental and theoretical study of cobalt clusters $Co_n^{±/0}$ in reacting with water and oxygen. All the $Co_n^{±/0}$ clusters were found to react with oxygen regardless of the presence of water or not. However, water dissociation is observed only for the anionic $Co_n^-$ ($n = 5$–59) clusters, but the cationic $Co_n^+$ and neutral $Co_n$ clusters do not support the observation of dehydrogenation except for $Co_3^+$. Combined with the DFT results, we unveil the bonding mechanisms of the $Co_n^{±/0}$ clusters and illustrate the reaction kinetics of typical $Co_n^-$ clusters toward water in forming $Co_nO^-$, $Co_n(OH)_2^-$, and $Co_nO_2^-$ products. Notably, the cobalt catalysis for dehydrogenation processes is not inhibited in the presence of oxygen; instead, a series of products of oxidation and partial dehydrogenation embody the corrosion of nano-cobalt surfaces.

## Methods

### Experimental methods

The instrumentation used in this study is based on a customised reflection time-of-flight mass spectrometer (Re-TOFMS). Detailed descriptions can be found in our previous publications[33,35,44]. In brief, the Re-TOFMS is equipped with a flow tube reactor which is connected with dual pulse valves enabling reactions with two reactant gases (e.g., $O_2$ and water). Isotopic chemicals of both $D_2O$ and $H_2^{18}O$ were used to help identify the reaction products. The $Co_n^±$ clusters were prepared by ablating a clean cobalt disk ($\Phi = 16$ mm, 99.95%) with a pulsed laser (10 Hz 532 nm Nd: YAG) in the presence of helium buffer gas (99.999%, 10.0 atm). The $Co_n^±$ clusters were prepared and ejected out of a nozzle ($\Phi = 2$ mm, $L = 35$ mm) during a supersonic expansion process controlled by the pulsed valve (Series 9, General Valve). For reactions between the cobalt clusters and water ($D_2O$ and $H_2^{18}O$), water vapour was injected into the flow tube reactor ($\Phi = 6$ mm, $L = 60$ mm) by the He (99.999%, 1 atm) bubbling method. Oxygen reactant

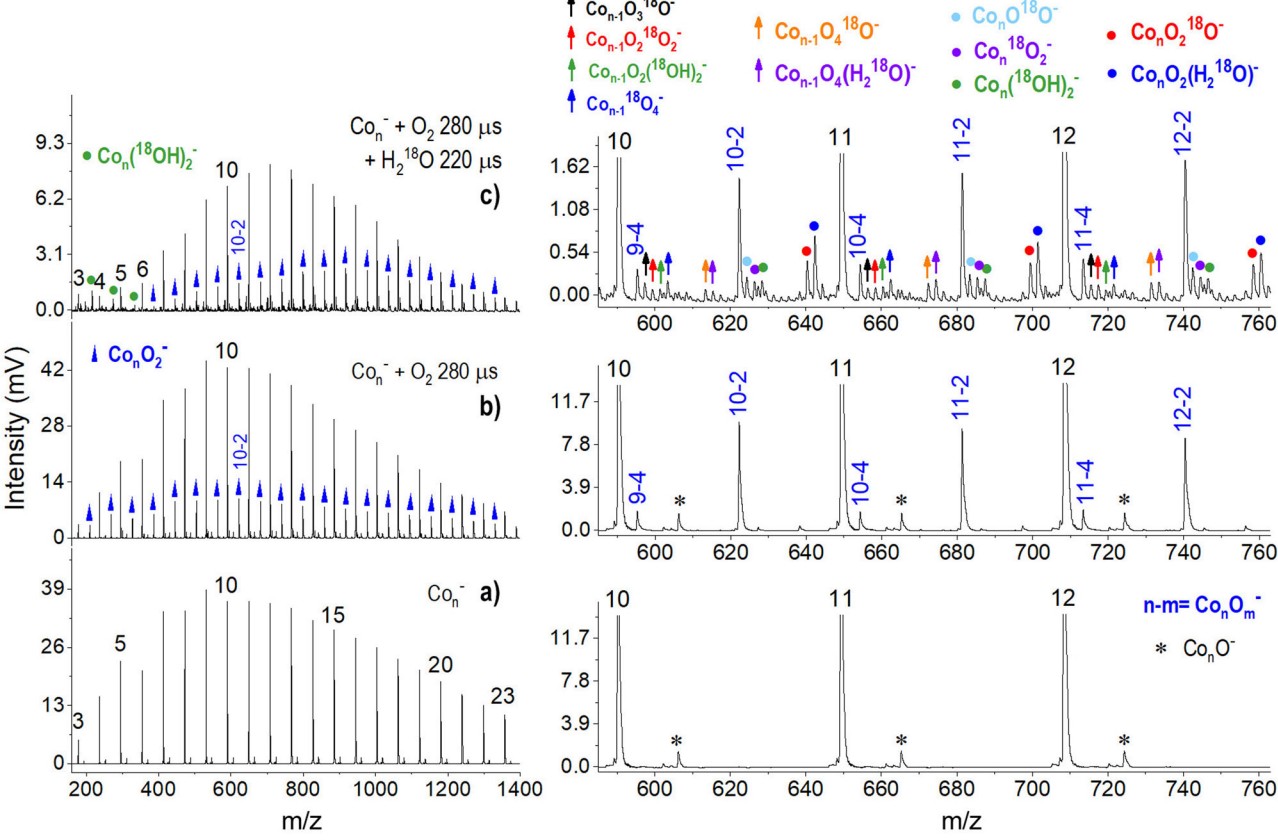

**Fig. 6 | Co_n⁻ reacting with O₂ and D₂O.** **a** Typical mass spectrum of the anionic Co_n⁻ ($n = 3$–23) clusters. **b** The mass spectrum after their reactions with oxygen (1% in He). **c** The mass spectrum after reactions with both O₂ and H₂¹⁸O in the flow tube, controlled by two pulse valves. The enlarged mass ranges for Co₁₀₋₁₂⁻ are given on the right side, respectively.

was diluted (1% in helium) and introduced from the other pulse valve connected to the same flow tube reactor. The reactants were controlled by varying the on-time pulse width. All metal clusters and their reaction products were detected and analysed by the Re-TOFMS. For the neutral $Co_n$ clusters reacting with $D_2O$, we used an all-solid-state deep ultraviolet (DUV) laser (177.3 nm wavelength, 15.5 ps pulse width, 10 Hz repletion rate, and ~15 μJ energy per pulse) with a head-to-head mode in the ionisation zone.

## DFT calculation methods

The DFT calculations were performed with the PBE-D3 corrected functional[45] using the Gaussian 16 programme[46]. The geometric optimisation and reaction coordinate research were carried out using the balanced triple-zeta def2-TZVP basis set[47] for Co, O and H atoms. Vibrational frequency calculations were carried out to ensure that the lowest-energy structures of reaction products have no imaginary frequencies and the transition states (TSs) have only one imaginary frequency. All energies were corrected with zero-point vibrations and the intrinsic reaction coordinate (IRC) scan was employed to ensure a connection with both intermediates in the reaction pathway. The natural bond orbital (NBO), electrostatic potential (ESP), and charge decomposition analysis were analysed by Multiwfn software[48]. Orbitals and ESP patterns were drawn by the visual molecular dynamics (VMD) software[49].

## Data availability

The data that support the findings of this study are available within the article and its Supplementary Information or from the corresponding author upon reasonable request.

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

## Acknowledgements

This work was financially supported by the National Natural Science Foundation of China (92261113 and 21722308) and the Key Project of Frontier Science Research of the Chinese Academy of Sciences (QYZDB-SSW-SLH024).

## Author contributions

L.G. and S.L. conducted the experiments. L.G., P.W. and R. S. contributed to the theoretical calculations and analyses. Z.L. and J.Z. designed this project. All authors contributed to analysing the data and writing the manuscript.

## Competing interests

The authors declare no competing interests.
