## [Peer Review File · Communications Chemistry]

Reviewers' comments:

Reviewer #1 (Remarks to the Author):

Report

The manuscript (ID - COMMSCHEM-23-0642) titled as, "The reactivity of $\text{Co}_n\pm/O$ clusters with water and oxygen: On the nature of corrosion of cobalt nanometals" is submitted by Luo and colleagues have described the reactivity of gas phase cobalt clusters with water and oxygen.

The manuscript is written properly. However, there are several sections in the manuscript that are not explained with proper justification. The manuscript is not recommended for publication in the Communications Chemistry in the current form.

Some of the comments and questions are given below.

1. $\text{Co}_n\pm/O$ ($n = 1-30$) clusters are not metals. Their metallicity was not proven. Therefore, the title should be changed.
2. A simplified schematic of instrument/experimental setup should be included with the Figure 1.
3. Corrosion is not an appropriate terminology for this type of controlled reaction. Here the reaction temperature and pressure are significantly different from the ambient corrosion condition.
4. Why did the authors termed cobalt clusters reaction with H_2O molecules that water splitting? Cobalt clusters reacted with water molecules and produced cobalt oxides which are detected. It did not produce hydrogen and oxygen from water.
5. Please revise Figure 2 caption.
6. In all the mass spectra in the manuscript, abscissa should be uniform. In Figure 3, it is given as 'm/z' whereas these are given as 'amu'.
7. Can the authors comment how neutral cobalt clusters and their coordination complexes with D_2O are detected using mass spectrometry (Figure 3B)?

Reviewer #2 (Remarks to the Author):

The reactivity of $\text{Co}_n \pm/O$ clusters with water and oxygen: On the nature of corrosion of cobalt nanometals

The manuscript examines the reaction of water with neutral, cationic, and anionic Co nanoclusters. The manuscript analyzes the differences in the reaction pathway with the different charge states and different sizes of Co nanoclusters. Overall, the article provides insight into reactivity on Co nanoclusters which is critical for a variety of applications, specifically in catalysis. Additionally, in this size regime, the work allows for a smooth transition and understanding between bulk materials and molecular complexes.

Overall, considering the publication criteria in Comm Chem, I reject the current work, with notice that

additional revision may justify a resubmission. Overall, I think the conclusions provided in the current version are not technically sound and need further justification. Additionally, I find that the impact of the paper is missing. What in this work is exciting? What should we take from it? There is a bit of a disconnect between experiment and theory that would get fixed with further discussion.

The following comments provide reasoning for my decision.

- Considering that the DFT calculations have already been calculated with vibrational frequencies as stated in the computational details, I would suggest that the binding energies and energy diagrams be calculated with Gibbs Free Energy correction rather than E with ZPE (comp details) or E_{ad} (as outlined in figure S12). This will make the thermodynamic discussion of the mechanism more reliable.
- There needs to be a more direct reference of what the authors are considering as adsorbed in the calculations. Several small anionic clusters do not bind H₂O directly as shown in figure S9, and it does not look like the water molecule has changed in geometry. From this, I assume there is no interaction between the Co and H₂O water molecule. It could be that the molecule is physisorbed, but in that case, is it a result of the added dispersion correction?
- To the above point, I don't think there is interaction between Co₆⁻ and H₂O with a bond length > 3.2 (figure S13). Fixing this comment will prompt more discussion, which I think will greatly enhance the current work.
- The binding modes need to be clarified. Further, analyzing the structures, I do not see Co-H bonded species as reported in line 141 of the manuscript.
- The reaction mechanisms all in all need more discussion from a theoretical and experimental perspective. There are some cool things happening that just are not fully explained. There needs to be more agreement. One example is that the manuscript claims that anionic clusters (n=5-59) are the only ones that are responsible for water splitting (lines 224-225). But with the theory in figure 4 it looks like water splitting is happening in all of the clusters.
- In figure 4 ESP is shown in the inset. In figure 5, HOMO is shown in the inset, but these are not discussed at all in the text. What does this show? Why are they important?
- These are very small clusters, so a higher level of theory would be feasible. What is the reasoning behind the choice of theory (Specifically XC functional)? Was charge transfer considered between the water and Co? How many conformations were tried? Where were the H₂O molecules bonded initially to the Co clusters?
- Further, what is the reasoning in choosing Co₆ system for comparison between the charge states? Can more connection be made here between theory and experiment?
- Discussion of thermodynamics and kinetics needs to be fine-tuned. What does kinetically favorable mean? In comparison to what?
- Spin multiplicities for figure S7, S8 and S9 are currently missing, but I think they would greatly add to the discussion in lines 155-164.
- Is mass spec enough of characterization for these reactions? Have the experiments considered follow up experiments with IR such as demonstrated in the following article?
Kiawi, D.M. et al. Water Adsorption on Free Cobalt Cluster Cations. *The Journal of Physical Chemistry A* 2015, 119 (44), 10828-10837
- Overall, I think there needs to be more discussion in the introduction. It is stated multiple times that Co bulk is not catalytically active for water splitting, however, there are several Co complexes that are active.

I would think these small nanoclusters are closer to complexes than the bulk material. Hence, I think this discussion should be expanded (similar to what was done with Al and VnO but in direct terms of Co complexes and water splitting)

- Additionally in the introduction, I am unclear with what is being said in lines 67 and 68. Can the authors explain further how it is being restricted rather than just saying it does not agree with the principles of the activity series?
- How was the corrosion equation obtained? Lines 201-203, I like the discussion afterwards, but I am unsure of how the resulting equation came about. Would this change for neutral and cationic clusters? If so, how?
- The authors should add a few references to the statement in line 206 with Au and Pt.

Reviewer #3 (Remarks to the Author):

Title: The reactivity of $\text{Co}_n\pm/0$ clusters with water and oxygen: On the nature of corrosion of cobalt nanometals

Manuscript ID: COMMSCHEM-23-0642

Geng et al. have reported corrosive nature of cobalt clusters of different nuclearity and charge state. Although the work is performed in detail, the work lacks novelty. I have the following questions:

1. Metals are inherently more reactive in the nanoscale, for example Gold. How this finding is different?
2. In Figure 3B, the authors have mentioned the clusters as neutral. How they were detected by MS?
3. Figure 2, the X axis is labelled as Mass (amu). Please change to m/z.
4. The authors mentioned the reaction with H₂O as water splitting. This needs justification.
5. The reactivity with O₂ and H₂O simply cannot be termed as corrosion.
6. The authors mentioned the Co_n clusters as nanometals. How the metallicity appeared at such small size?
7. What is the proof for H₂ release?
8. "Interestingly, the lowest-energy structures of water-adsorption on the cationic and neutral clusters prefer Co-O coordination (i.e., forming [Co_nOH₂]^{+/0}); however, water adsorption on the Co_n⁻ clusters results in Co-H bonding (Co_n-•••H-OH)." Is there any experimental proof for this statement?

Response to the Reviewers' Comments

Reviewer: 1

The manuscript (ID - COMMSCHEM-23-0642) titled as, "The reactivity of $\text{Co}_n^{\pm/0}$ clusters with water and oxygen: On the nature of corrosion of cobalt nanometals" is submitted by Luo and colleagues have described the reactivity of gas phase cobalt clusters with water and oxygen.

The manuscript is written properly. However, there are several sections in the manuscript that are not explained with proper justification. The manuscript is not recommended for publication in the Communications Chemistry in the current form.

Some of the comments and questions are given below.

【Re:】 We thank the referee for the summary and comments on this manuscript. By fully addressing all the following comments, we have made major revisions and conducted a double check throughout the manuscript.

1. $\text{Co}_n^{\pm/0}$ ($n = 1-30$) clusters are not metals. Their metallicity was not proven. Therefore, the title should be changed.

【Re:】 We have revised it into "On the nature of $\text{Co}_n^{\pm/0}$ clusters reacting with water and oxygen".

2. A simplified schematic of instrument/experimental setup should be included with the Figure 1.

【Re:】 We have added a sketch of the equipment in Figure S1 of Supporting Information.

3. Corrosion is not an appropriate terminology for this type of controlled reaction. Here the reaction temperature and pressure are significantly different from the ambient corrosion condition.

【Re:】 By taking the referee's advice, we have reworded the "corrosion" in the context.

4. Why did the authors termed cobalt clusters reaction with H_2O molecules that water splitting? Cobalt clusters reacted with water molecules and produced cobalt oxides which are detected. It did not produce hydrogen and oxygen from water.

【Re:】 We understand the referee's comment. In addition to the known water splitting by electrolysis and photocatalysis, the terminology "water splitting" is also used in cluster science (*Science* 2009, 323, 492; *Coord. Chem. Rev.* 2022, 457, 214419) for the strategies without photocatalysis. While we cannot see H_2 production in the mass spectrometry, evidence for H_2 release by water molecules reacting with Co_n^- clusters can be inferred by the observation of a series of products $[\text{Co}_n^{18}\text{O}]^-$, $[\text{Co}_n^{18}\text{O}_2]^-$ and $[\text{Co}_n(^{18}\text{OH})_2]^-$. By taking the referee's comment, we have changed "water splitting" to "water dissociation".

5. Please revise Figure 2 caption.

【Re:】 We have corrected it.

6. In all the mass spectra in the manuscript, abscissa should be uniform. In Figure 3, it is given as 'm/z' whereas these are given as 'amu'.

【Re:】 We have used m/z for all.

7. Can the authors comment how neutral cobalt clusters and their coordination complexes with D₂O are detected using mass spectrometry (Figure 3B)?

【Re:】 We thank the referee for pointing out this, we have added the description in Experimental methods.

“For the neutral Co_n clusters reacting with D₂O, we used an all-solid-state deep ultraviolet ionization (DUV) laser (177.3 nm wave length, 15.5 ps pulse width, 10 Hz repetition rate, and ~15 μJ energy per pulse) with a head-to-head mode in the ionization zone.”

Reviewer: 2

The reactivity of Co_n^{±/0} clusters with water and oxygen: On the nature of corrosion of cobalt nanometals

The manuscript examines the reaction of water with neutral, cationic, and anionic Co nanoclusters. The manuscript analyzes the differences in the reaction pathway with the different charge states and different sizes of Co nanoclusters. Overall, the article provides insight into reactivity on Co nanoclusters which is critical for a variety of applications, specifically in catalysis. Additionally, in this size regime, the work allows for a smooth transition and understanding between bulk materials and molecular complexes. Overall, considering the publication criteria in Comm Chem, I reject the current work, with notice that additional revision may justify a resubmission. Overall, I think the conclusions provided in the current version are not technically sound and need further justification. Additionally, I find that the impact of the paper is missing. What in this work is exciting? What should we take from it? There is a bit of a disconnect between experiment and theory that would get fixed with further discussion. The following comments provide reasoning for my decision.

【Re:】 We thank the referee for the summary and comments on this manuscript. Cobalt, in contrast to aluminium and vanadium, does not support water dehydrogenation at room temperature according to the activity series of metals, but cobalt nanocatalysts evoke great interest. It is important to unveil cobalt clusters' reactivity of both oxidation and water dehydrogenation, thus to better understand the corrosion mechanism of cobalt nanometals. By fully addressing all the following comments, we have made major revisions and conducted a double check throughout the manuscript.

1. Considering that the DFT calculations have already been calculated with vibrational frequencies as stated in the computational details, I would suggest that the binding energies

and energy diagrams be calculated with Gibbs Free Energy correction rather than E with ZPE (comp details) or Ead (as outlined in figure S12). This will make the thermodynamic discussion of the mechanism more reliable.

【Re:】 We have added the binding energies and energy diagrams with Gibbs Free Energy correction in Tables S2 and S3 and Figure S20. The relative energy trend after Gibbs free energy correction is similar to the ZPV-corrected energies, although the relative Gibbs free energies bring forth minor difference of a few transition states. Notably, for the gas-phase cluster reactions, the two energies are widely used in the literature; somebody else may believe the ZPV-corrected total energies reflect the actual situation of supersonic expansion of molecular beams and often find well consistence of experimental and theoretical results.

2. There needs to be a more direct reference of what the authors are considering as adsorbed in the calculations. Several small anionic clusters do not bind H₂O directly as shown in figure S9, and it does not look like the water molecule has changed in geometry. From this, I assume there is no interaction between the Co and H₂O water molecule. It could be that the molecule is physisorbed, but in that case, is it a result of the added dispersion correction?

【Re:】 Yes, there is similar situation for H₂O adsorption on the anionic aluminum clusters (*J. Phys. Chem. A* 2008, 112, 1313). For cobalt clusters in this work, Co₁₋₆⁻·H₂O prefer Co-H bonding (Co_n⁻···H-OH) after the optimization even for an initial structure of Co-OH₂. Compared with the results by PBE-D3/def2-TZVP, we have checked the results by PBE/def2-tzvp without dispersion correction. The results calculated by the two methods are similar to each other, as shown below.

Fig. # Ground state structures of the Co_n⁻·H₂O (n = 1–6) clusters, optimized by PBE/def2-tzvp (Left) and PBE-D3/def2-TZVP (Right) respectively. Letter M refers to spin multiplicities.

3. To the above point, I don't think there is interaction between Co₆⁻ and H₂O with a bond length > 3.2 (figure S13). Fixing this comment will prompt more discussion, which I think will greatly enhance the current work.

【Re:】 Although the Co-O bond length in the Co₆⁻·H₂O cluster is larger than 3.2, the Co-H bond length is 2.66. Notably, the binding energy of Co₆⁻ with H₂O (0.52 eV) is larger

than the normal hydrogen bond (5-10 kcal/mol, i.e., 0.21-0.42 eV), indicating that Co_6^- does interact with water molecules.

4. The binding modes need to be clarified. Further, analyzing the structures, I do not see Co-H bonded species as reported in line 141 of the manuscript.

【Re:】 We have updated the Figure S10 of Co-H bonded species, and also rewritten the manuscript as below,

“...water adsorption on the small Co_n^- ($n = 1,2,3,5,6$) clusters could result in Co-H bonding ($\text{Co}_n^- \cdots \text{H-OH}$).”

5. The reaction mechanisms all in all need more discussion from a theoretical and experimental perspective. There are some cool things happening that just are not fully explained. There needs to be more agreement. One example is that the manuscript claims that anionic clusters ($n=5-9$) are the only ones that are responsible for water splitting (lines 224-225). But with the theory in figure 4 it looks like water splitting is happening in all of the clusters.

【Re:】 The referee is right. The cool He buffer gas could take away part of the energy during the reaction, rendering the cationic and neutral clusters not having enough energy for dehydrogenation especially those with large energy barriers. We have added this discussion in the context,

“Notably, the cool He buffer gas could take away part of the energy during the reaction, rendering the cationic and neutral clusters not having enough energy for dehydrogenation especially for those having large energy barriers of the transition states.”

6. In figure 4 ESP is shown in the inset. In figure 5, HOMO is shown in the inset, but these are not discussed at all in the text. What does this show? Why are they important?

【Re:】 We thank the referee for pointing this out. The ESP and LUMO/HOMO patterns are shown to help understand the favorable orientation and active sites for H_2O adsorption. We have added this information in the figure caption respectively,

“The inset shows electrostatic potentials (EPS, $\text{kcal}\cdot\text{mol}^{-1}$) for a H_2O molecule in approaching the Co_6^- cluster with spontaneously regulated orientation.”

“The insets show the corresponding structures. The inset on the left bottom shows the HOMO and LUMO patterns of $[\text{Co}_{11}\cdot\text{H}_2\text{O}]^-$.”

7. These are very small clusters, so a higher level of theory would be feasible. What is the reasoning behind the choice of theory (Specifically XC functional)? Was charge transfer considered between the water and Co? How many conformations were tried? Where were the H_2O molecules bonded initially to the Co clusters?

【Re:】 Yes, we calculated anionic, neutral, and cationic cobalt clusters of 1-13 atoms with varied spin states, adsorption structures, and reaction pathways. We chose the widely used pure PBE without XC functional because it is relatively efficient and accurate for energetics

calculations and widely used in commercial codes. Also, we added dispersion corrections considering the presence of weak adsorption of water molecules. The charge transfer was considered in Figure S13 in the Supporting Information; a typical example below shows the typical isomers of the neutral $\text{Co}_n \cdot \text{H}_2\text{O}$ ($n = 1-13$) clusters.

Fig. # Isomers of the neutral $\text{Co}_n \cdot \text{H}_2\text{O}$ ($n = 1-13$) clusters, optimized by PBE-D3/def2-TZVP. Letter M refers to spin multiplicities. The relative energy are given in eV.

8. Further, what is the reasoning in choosing Co_6 system for comparison between the charge states? Can more connection be made here between theory and experiment?

【Re:】 The $\text{Co}_6^{\pm/0}$ is chosen for comparison because the geometric structures of neutral, cationic and anionic are identical to each other, and the highly symmetric octahedron reduce the diversity of surface active sites.

9. Discussion of thermodynamics and kinetics needs to be fine-tuned. What does kinetically favorable mean? In comparison to what?

【 Re: 】 Thermodynamically favorable but/and kinetically (un)favorable usually correspond to the reaction energy and the reaction transition states (Thermodynamics and Kinetics (stanford.edu)). Thermodynamically favorable usually refers to exothermic reactions; while kinetically favorable means the transition states have surmountable energy barriers. We have tried to reword the expressions in the context.

10. Spin multiplicities for figure S7, S8 and S9 are currently missing, but I think they would greatly add to the discussion in lines 155-164.

【Re:】 We have added the spin multiplicities for Figures S8, S9 and S10.

11. Is mass spec enough of characterization for these reactions? Have the experiments considered follow up experiments with IR such as demonstrated in the following article? Kiawi, D.M. et al. Water Adsorption on Free Cobalt Cluster Cations. The Journal of Physical Chemistry A 2015, 119 (44), 10828-10837

【Re:】 Mass spectrometry is widely used in cluster chemistry for reaction analysis (*Chem. Rev.* 2016, 116, 14456). The IRMPD is helpful to identify cluster structures (as in some colleagues' lab and we ourselves have it), but we find difficulty to apply such an action spectroscopy to characterize gas-phase reactions.

12. Overall, I think there needs to be more discussion in the introduction. It is stated multiple times that Co bulk is not catalytically active for water splitting, however, there are several Co complexes that are active. I would think these small nanoclusters are closer to complexes than the bulk material. Hence, I think this discussion should be expanded (similar to what was done with Al and VnO but in direct terms of Co complexes and water splitting)

【Re:】 We have tried to improve the Introduction. But, Co complexes, ligand-protected metal nanoclusters, and naked Co clusters are essentially different.

13. Additionally in the introduction, I am unclear with what is being said in lines 67 and 68. Can the authors explain further how it is being restricted rather than just saying it does not agree with the principles of the activity series?

【Re:】 We have added a reference to this, seen as below.

Metals	Metal Ion	Reactivity
K	K ⁺	reacts with water
Na	Na ⁺	reacts with water
Li	Li ⁺	reacts with water
Ba	Ba ²⁺	reacts with water
Sr	Sr ²⁺	reacts with water
Ca	Ca ²⁺	reacts with water
Mg	Mg ²⁺	reacts with acids
Al	Al ³⁺	reacts with acids
Mn	Mn ²⁺	reacts with acids
Zn	Zn ²⁺	reacts with acids
Cr	Cr ²⁺	reacts with acids
Fe	Fe ²⁺	reacts with acids
Cd	Cd ²⁺	reacts with acids
Co	Co ²⁺	reacts with acids
Ni	Ni ²⁺	reacts with acids
Sn	Sn ²⁺	reacts with acids

14. How was the corrosion equation obtained? Lines 201-203, I like the discussion afterwards, but I am unsure of how the resulting equation came about. Would this change for neutral and cationic clusters? If so, how?

【Re:】 The equation was written based on the reactants and the products observed in the mass spectra. Since it is demonstrated that the reaction of cobalt clusters with water is charge-dependent in experimental results, the neutral clusters have the weakest adsorption activity and do not undergo similar results. In comparison, the cations show strong adsorption of water but a weak reactivity to dissociate water.

15. The authors should add a few references to the statement in line 206 with Au and Pt.

【Re:】 We have referred to some literature and add one in the reference list (Ref.43).

Reviewer: 3

Title: The reactivity of $\text{Co}_n^{\pm/0}$ clusters with water and oxygen: On the nature of corrosion of cobalt nanometals

Manuscript ID: COMMSCHEM-23-0642

Geng et al. have reported corrosive nature of cobalt clusters of different nuclearity and charge state. Although the work is performed in detail, the work lacks novelty.

【Re:】 We thank the referee for the simple summary and comments on this manuscript. By fully addressing all the following comments, we have made major revisions and conducted a double check throughout the manuscript.

I have the following questions:

1. Metals are inherently more reactive in the nanoscale, for example Gold. How this finding is different?

【Re:】 Nanometals such as gold are generally more reactive than bulk metals because of higher surface energy, and enhanced specific surface area. This finding on the reactions of $\text{Co}_n^{\pm/0}$ clusters dig deep into subnanometals with fine structure at atomic precision and different charge states.

2. In Figure 3B, the authors have mentioned the clusters as neutral. How they were detected by MS?

【Re:】 We are sorry to not describe this clearly. We have added a sketch drawing of the instrumentation (newly added Figure S1) and more description in Experimental methods.

“For the neutral Co_n clusters reacting with D_2O , we used an all-solid-state deep ultraviolet (DUV) laser (177.3 nm wave length, 15.5 ps pulse width, 10 Hz repetition rate, and $\sim 15 \mu\text{J}$ energy per pulse) with a head-to-head mode in the ionization zone.”

3. Figure 2, the X axis is labelled as Mass (amu). Please change to m/z.

【Re:】 We have fixed it.

4. The authors mentioned the reaction with H₂O as water splitting. This needs justification.

【Re:】 We have revised “water splitting” to “water dissociation”.

5. The reactivity with O₂ and H₂O simply cannot be termed as corrosion.

【Re:】 We have rephrased the related expression in the context including the title.

6. The authors mentioned the Co_n clusters as nanometals. How the metallicity appeared at such small size?

【Re:】 We classify these naked Co_n clusters as nanometals and they do have comparable electron ionization energies, seen as below. It could be not suitable to evaluate their metallicity like large nanoparticles; instead, superatomic features are applied in this field.

Figure # The ionization energies (VIE) of Co_n clusters (*Natl. Sci. Rev.* 2021, 8, nwa201).

7. What is the proof for H₂ release?

【Re:】 H₂ release is inferred by observing the products [Co_n¹⁸O]⁻, [Co_n¹⁸O₂]⁻ and [Co_n(¹⁸OH)₂]⁻ in the mass spectra.

8. “Interestingly, the lowest-energy structures of water-adsorption on the cationic and neutral clusters prefer Co-O coordination (i.e., forming [Co_nOH₂]⁺⁰); however, water adsorption on the Co_n⁻ clusters results in Co-H bonding (Co_n⁻⋯H-OH).” Is there any experimental proof for this statement?

【Re:】 We thank the referee for this remark and suggestion. Sorry but we are not able to provide any experimental evidence for the reaction intermediates of Co-H bonding (Co_n⁻⋯H-OH) at present. Our discussion is based on the first-principles calculations.

Again, we thank all the referees for their informative comments which help us to have improved this manuscript significantly.

REVIEWERS' COMMENTS:

Reviewer #1 (Remarks to the Author):

The revised manuscript (ID: COMMSCHEM-23-0642A), titled as "On the nature of Con \pm /0 clusters reacting with water and oxygen" by Luo and colleagues has shown substantial improvement in the quality. They have addressed all the questions with proper justification.

I now recommend the manuscript for the publication. Further review is not required.

Reviewer #2 (Remarks to the Author):

Overall, I think the points raised in the first round of revision have been sufficiently addressed and have added depth and understanding to the manuscript.

I apologize for the bad wording in the initial revision. Toward comments 9 and 13, I understand what the terms mean, but was intending for explanation in terms of the project rather than 'general' terms. Toward this point, however, I think the general terms being outlined more directly have enhanced understanding from a reader's perspective and have improved the manuscript.

Reviewer #3 (Remarks to the Author):

The authors now have answered my queries and modified the manuscript accordingly. The manuscript may be accepted in the present form.